# Effect of moderate elevated intra-abdominal pressure on lung mechanics and histological lung injury at different positive end-expiratory pressures

Mascha O. Fiedler[1], B. Luise Deutsch[2], Emilis Simeliunas[1], Dovile Diktanaite[1], Alexander Harms[3], Maik Brune[4], Florian Uhle[1], Markus Weigand[1], Thorsten Brenner[1], Armin Kalenka[5,6]*

1 Department of Anesthesiology, Heidelberg University Hospital, Heidelberg, Germany, 2 Justus-Liebig-University, Faculty of Medicine, Giessen, Germany, 3 Heidelberg University Hospital, Institute of Pathology, Heidelberg, Germany, 4 Department of Internal Medicine I and Clinical Chemistry, Heidelberg University Hospital, Heidelberg, Germany, 5 Department of Anesthesiology and Intensive Care Medicine, Hospital Bergstrasse, Heppenheim, Germany, 6 Faculty of Medicine, University of Heidelberg, Heidelberg, Germany

* armin.kalenka@med.uni-heidelberg.de

**Data Availability Statement:** All relevant data are within the manuscript and its Supporting Information files.

## Abstract

### Introduction

Intra-abdominal hypertension (IAH) is a well-known phenomenon in critically ill patients. Effects of a moderately elevated intra-abdominal pressure (IAP) on lung mechanics are still not fully analyzed. Moreover, the optimal positive end-expiratory pressure (PEEP) in elevated IAP is unclear.

### Methods

We investigated changes in lung mechanics and transformation in histological lung patterns using three different PEEP levels in eighteen deeply anesthetized pigs with an IAP of 10 mmHg. After establishing the intra-abdominal pressure, we randomized the animals into 3 groups. Each of n = 6 (Group A = PEEP 5, B = PEEP 10 and C = PEEP 15 cmH$_2$O). End-expiratory lung volume (EELV/kg body weight (bw)), pulmonary compliance (C$_{stat}$), driving pressure (ΔP) and transpulmonary pressure (ΔP$_L$) were measured for 6 hours. Additionally, the histological lung injury score was calculated.

### Results

Comparing hours 0 and 6 in group A, there was a decrease of EELV/kg (27±2 vs. 16±1 ml/kg; p<0.05) and of C$_{stat}$ (42±2 vs. 27±1 ml/cmH$_2$O; p<0.05) and an increase of ΔP (11±0 vs. 17±1 cmH$_2$O; p<0.05) and ΔP$_L$ (6±0 vs. 10±1 cmH$_2$O; p<0.05). In group B, there was no significant change in EELV/kg (27±3 vs. 24±3 ml/kg), but a decrease in C$_{stat}$ (42±3 vs. 32±1 ml/cmH$_2$0; p<0.05) and an increase in ΔP (11±1 vs. 15±1 cmH$_2$O; p<0.05) and ΔP$_L$ (5±1 vs. 7±0 cmH$_2$O; p<0.05). In group C, there were no significant changes in EELV/kg (27±2 vs. 29±3 ml/kg), ΔP (10±1 vs. 12±1 cmH$_2$O) and ΔP$_L$ (5±1 vs. 7±1 cmH$_2$O), but a significant

**Funding:** The author(s) received no specific funding for this work.

**Competing interests:** The authors have declared that no competing interests exist.

decrease of $C_{stat}$ (43±1 vs. 37±1 ml/cmH$_2$O; p<0.05). Histological lung injury score was lowest in group B.

## Conclusions

A moderate elevated IAP of 10 mmHg leads to relevant changes in lung mechanics during mechanical ventilation. In our study, a PEEP of 10 cmH$_2$O was associated with a lower lung injury score and was able to overcome the IAP induced alterations of EELV.

## Introduction

The interactions between the abdominal and the thoracic compartments represent a challenge for ICU physicians [1, 2]. Approximately 50% of intra-abdominal pressure (IAP) is transmitted to the intrathoracic compartment [3–6]. It therefore has a direct impact on functional residual capacity (FRC), end-expiratory lung volume (EELV), driving pressure (ΔP) and transpulmonary pressure (ΔP$_L$). Almost half of the patients admitted to the ICUs worldwide develop intra-abdominal hypertension (IAH). Two-thirds of these cases were already present on the day of ICU admission [7]. An elevated IAP can be classified as follows: 1. intra-abdominal hypertension (IAH) with an IAP above 12 mmHg and 2. abdominal compartment syndrome with an IAP above 20 mmHg [8, 9]. Both are an independent risk factor for organ failure and mortality in the ICU [1, 7].

Clinical studies on critically ill patients identified an average IAP of 10 mmHg in supine position [1, 10, 11]. Moderate elevated IAP also occurs in obesity, pregnancy and during anesthesia [12–14]. The role of this moderate elevated IAP on lung mechanics and potential organ failure is yet not fully analysed.

We therefore analysed the effect of a moderate elevated IAP of 10 mmHg on lung mechanics in a porcine model up to 6 hours. To investigate the consequences of different PEEP levels, we used three levels of PEEP (5, 10, 15 cmH$_2$O). The hypothesis in our study was that a PEEP of 10 cmH$_2$O in moderate elevated IAP (10 mmHg) is protective by reducing lung injury and preserving the EELV during mechanical ventilation.

## Materials and methods

### Animal preparation and instrumentation

The protocol was approved by the responsible committee for animal research (Regierungspräsidium Karlsruhe, No. 35–9185.81/G-161/17). The animals were kept within the interfacultary biomedical faculty of the university of Heidelberg and were provided by a local pig breeder. All proceedings were in accordance with animal welfare notes regulated by German law. After overnight fasting with free access to water, 18 female domestic pigs were anaesthetized intramuscularly in combination with 7 mg/kg Azaperon (Stresnil$^®$, Lilly, Bad Homburg, Germany), 8 mg/kg Ketaminhydrochlorid 10% (Ketamin10%$^®$, Bremer Pharma, Warburg, Germany) and 0.3 mg/kg Midazolam (Midazolam, Hameln Pharma, Hameln, Germany). Anaesthesia was maintained by continuous infusion of 6 mg/kg/h Ketanest S (Pfizer Pharma, Berlin, Germany), 3.6 mg/kg/h Midazolam and 10–30 mg/kg/h Propofol 2% (Propofol, Fresenius Kabi, Bad Homburg, Germany). There was no use of neuromuscular blockers. Adequacy of the depth of anaesthesia was regularly assessed by absence of spontaneous breathing efforts and lack of muscle tone.

After induction of anaesthesia the pigs were tracheotomised and ventilated with an intensive care ventilator (Carescape R860, GE Healthcare, Madison, USA) using an inspiratory oxygen concentration ($F_iO_2$) of 0.4 in a pressure-controlled mode with volume guaranty. Also, a tidal volume of 8 ml/kg body weight, an inspiration/expiration ratio of 1:2 and a PEEP of 5 $cmH_2O$ was provided.

A 5F thermistor-tipped catheter (PiCCO®, Pulsion Medical systems, Feldkirchen, Germany) and a central venous catheter (Logicath, Smiths Medical, Grasbrunn, Germany) were inserted with ultrasound guidance. Crystalloid solution (Sterofundin ISO, Braun, Melsungen, Germany) was infused to keep the study population hemodynamically stable during the experiment. A polyethylene catheter (Nutrivent multifunction nasogastric catheter, Sidam, San Giacomo Roncole, Italy) was used to measure esophageal pressure. Appropriate catheter position was confirmed as previously described [15].

After a midline laparotomy a large intra-abdominal balloon (200-litre weather balloon, Stratoflight, Blomberg, Germany) was placed in the peritoneal cavity. Correct position in all abdominal quadrants was ensured by visual inspection and partial inflation. The abdomen was carefully closed. A urine catheter was placed in the bladder.

## Measurements and calculations

Peak inspiratory airway pressure ($P_{Insp}$), PEEP, inspiratory esophageal pressure ($P_{EsInsp}$) and end-expiratory esophageal pressure ($P_{EsExp}$) were recorded from the ventilator. $\Delta P$ and $\Delta P_L$ were calculated as previously described [16]. Transpulmonary inspiratory pressure ($TPP_{Insp}$) was calculated as $TTP_{Insp} = P_{Insp} - P_{EsInsp}$ and transpulmonary expiratory pressure ($TPP_{Exp}$) as $TPP_{Exp} = PEEP - P_{EsExp}$. $C_{Stat}$ was measured by the ventilator during an inspiratory hold. Elastance of the respiratory system ($E_{RS}$) was calculated as $E_{RS} = (PInsp - PEEP) / V_T$, chest wall elastance ($E_{CW}$) as $E_{CW} = (PEsInsp - P_{EsExp}) / V_T$ and elastance of the lung ($E_L$) as $E_L = E_{RS} - E_{CW}$.

We measured EELV bedside as previously described [17] without interrupting mechanical ventilation on the designated PEEP level. CI (cardiac index) was calculated with the PiCCO® System. End-expiratory IAP ($IAP_{Endex}$) was measured as recommended [18, 19] and zeroed at midaxillary level [20]. P/F ratio was calculated based on the ratio of partial arterial pressure of oxygen to $F_iO_2$.

## Experimental protocol

Data were assessed after a 30 minutes stabilization period (H0). The abdominal balloon was then filled with water up to an $IAP_{Endex}$ of 10 mmHg. We randomized into group A (n = 6) with a PEEP of 5, group B (n = 6) = PEEP 10 and group C (n = 6) = PEEP 15 $cmH_2O$ for 6 hours (H6) (Fig 1).

At the end of the experimental protocol, the pigs were euthanized with an intravenous bolus of 200 mg Propofol followed by 40mmol potassium chloride. We exposed the complete right lung and regional lung samples were extracted to evaluate wet-dry weight ratio and to perform histological examination.

## Histology

Samples from the anterior, medial and dorsal position of the medial lobe were selected and immediately fixed in formalin. After fixation, the tissue samples were dehydrated and embedded. The sections were stained with hematoxylin and eosin. A pathologist, blinded to the study variables, evaluated each sample histologically to determine a lung injury score. To quantify the extent of histologic lung injury the pathologist used a lung injury scoring system [21] (S1

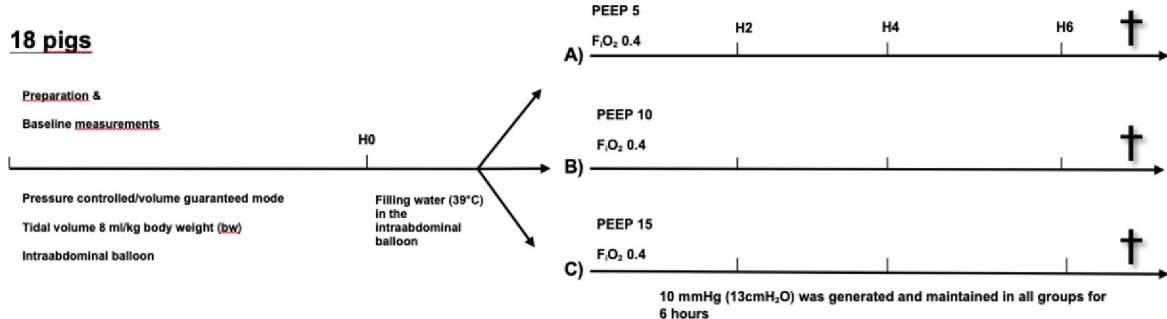

**Fig 1. Experimental model study protocol.** After instrumentation and a 30-min stabilization period animals underwent measurements at H0 with PEEP 5 and no elevated intraabdominal pressure (IAP). Thereafter the intraabdominal balloon was filled with water up to an IAP of 10 mmHg. Animals were randomized afterwards to the three different groups with group A: PEEP 5 cmH$_2$O, group B: PEEP 10 cmH$_2$O and group C: PEEP 15 cmH$_2$O. The IAP was held at 10 mmHg for 6 hours.

Table). Five independent variables were scored to generate the lung injury score. The sum of each of the five independent variables were weighted according to the relevance for acute lung injury [21]. The resulting lung injury score ranges from 0 to 1. Zero represents minimal to no damage and 1 the worst damage possible.

## Wet-dry weight ratio

Wet-dry weight ratio was measured in samples of the medial lobe. Samples were weighed, dried and then weighted again.

## Statistical analysis

Sample size was calculated based on the expected alterations in EELV using data from previous studies performed in our lab.

Statistical analysis was performed using SPSS (Version 25). H0 values and lung injury score were analyzed using a one-way ANOVA. In case of significance a post hoc analyze with a correction for multiple tests were performed. We used ANCOVA for comparing H0 with H6 data with post hoc analysis. Data are expressed as mean ± SEM (standard error of the mean). For all tests, P ≤ 0.05 was considered statistically significant.

## Results

We included 18 animals in this study with a weight of 47 ± 1 kg. There were no significant differences at H0 among the experimental groups with the exeption of the significant higher heart rate of group B compared to group A.

### Lung mechanics, hemodynamic parameters and oxygenation

When compared with data at H0 we observed several alterations after 6 hours of mechanical ventilation (H6) (Table 1). In group A EELV (1323±95 vs. 774±67; p<0.05), EELV/kg bw (27 ±2 vs. 16±1 ml/kg; p<0.05) (Figs 2 and 3) and C$_{stat}$ (42±2 vs. 27±1 ml/cmH$_2$O; p<0.05) decreased (S1 Fig). The ΔP (11±0 vs. 17±1 cmH$_2$O; p<0.05) and ΔP$_L$ (6±0 vs. 10±1 cmH$_2$O; p<0.05) showed an increase (Figs 2 and 3). In group B EELV (1337±168 vs. 1159±140; p>0.05) and EELV/kg (27±3 vs. 24±3 ml/kg; p>0.05) were not influenced. Nonetheless a significant decrease in C$_{stat}$ (42±3 vs. 32±1 ml/cmH$_2$O; p<0.05) and an increase in ΔP (11±1 vs. 15±1 cmH$_2$O; p<0.05) and ΔP$_L$ (5±1 vs. 7±0 cmH$_2$O; p<0.05) could be observed. In group C EELV (1231±89 vs. 1310±135; p>0.05), EELV/kg (27±2 vs. 29±3 ml/kg; p>0.05), ΔP (10±1 vs.

**Table 1.  Pulmonary and hemodynamic parameters in the setting of 6 hours mechanical ventilation with an intraabdominal pressure of 10 mmHg.**

| | | group A | group B | group C |
|---|---|---|---|---|
| Weight (kg) | H0 | 48±1 | 49±1 | 45±1 |
| IAP (mmHg) | H0 | 3±0 | 2±1 | 2±0 |
| IAP (mmHg) | H6 | 10±0* | 10±0* | 10±0* |
| EELV | H0 | 1323±95 | 1337±168 | 1231±89 |
| EELV | H6 | 774±67* | 1159±140# | 1310±135$ |
| EELV/kg | H0 | 27±2 | 27±3 | 28±2 |
| EELV/kg | H6 | 16±1* | 24±3# | 29±3$$ |
| $\Delta P$ | H0 | 11±0 | 11±1 | 10±1 |
| $\Delta P$ | H6 | 17±1* | 15±1*# | 12±1$$ |
| $\Delta P_L$ | H0 | 6±0 | 5±0 | 5±1 |
| $\Delta P_L$ | H6 | 10±1* | 7±0* | 7±1 |
| $TPP_{Insp}$ | H0 | 7±1 | 6±1 | 6±1 |
| $TPP_{Insp}$ | H6 | 11±2* | 7±1 | 12±2* |
| $TPP_{Exp}$ | H0 | 1±1 | 1±0 | 1±1 |
| $TPP_{Exp}$ | H6 | 1±1 | 1±1 | 5±1 |
| $C_{Stat}$ | H0 | 42±2 | 42±3 | 45±2 |
| $C_{Stat}$ | H6 | 27±1* | 32±1*# | 37±1*$$ |
| $E_{RS}$ | H0 | 28±1 | 29±1 | 28±1 |
| $E_{RS}$ | H6 | 44±1* | 37±2*# | 32±1*$$ |
| $E_{CW}$ | H0 | 14±1 | 15±2 | 14±1 |
| $E_{CW}$ | H6 | 18±3 | 20±2 | 13±1 |
| $E_L$ | H0 | 14±1 | 13±1 | 13±2 |
| $E_L$ | H6 | 26±3* | 17±1* | 19±2 |
| HR | H0 | 79±9 | 113±9# | 85±9 |
| HR | H6 | 77±12 | 99±6 | 70±4 |
| MAP | H0 | 78±6 | 98±5 | 88±7 |
| MAP | H6 | 100±6* | 102±6 | 104±6 |
| P/F ratio | H0 | 452±25 | 425±20 | 505±24 |
| P/F ratio | H6 | 439±19 | 396±18 | 501±20$ |
| CI | H0 | 4.9±0.5 | 5.7±0.3 | 4.4±0.4 |
| CI | H6 | 4.3±0.4 | 4.9±0.3 | 3.6±0.2 |

EELV = end-expiratory lung volume (ml), EELV/kg = end-expiratory lung volume per kg bodyweight (ml/kg), $\Delta P$ = driving pressure ($cmH_2O$), $\Delta P_L$ = transpulmonary pressure ($cmH_2O$), $TPP_{Insp}$ = inspiratory transpulmonary pressure ($cmH_2O$), $TPP_{Exp}$ = expiratory transpulmonary pressure ($cmH_2O$), $C_{Stat}$ = static pulmonary compliance ($ml/cmH_2O$), $E_{RS}$ = Elastance of the respiratory system ($cmH_2O/ml$), $E_{CW}$ = Elastance of the chest wall ($cmH_2O/ml$), $E_L$ = lung elastance ($cmH_2O/ml$), HR = heart rate (beats/min), MAP = mean arterial pressure (mmHg), P/F ratio = ratio between arterial pressure of oxygen and inspired oxygen concentration (mmHg), CI = cardiac index (l/min/m$^2$)

* = $P<0.05$ HO vs. H6

# = $p<0.05$ group A vs. group B

$ = $P<0.05$ group A vs. group C

$$ = $p<0.05$ group B vs. group C.

12±1 $cmH_2O$; $p>0.05$) and $\Delta P_L$ (5±1 vs. 7±1 $cmH_2O$; $p>0.05$) where not changed but a significant decrease of $C_{stat}$ (43±1 vs. 37±1 $ml/cmH_2O$; $p<0.05$) was found. The $TPP_{Insp}$ were different from H0 in group A and C (A: 6 ± 1 vs. 10 ± 1 $cmH_2O$, C: 6 ± 1 vs. 12 ± 2 $cmH_2O$,

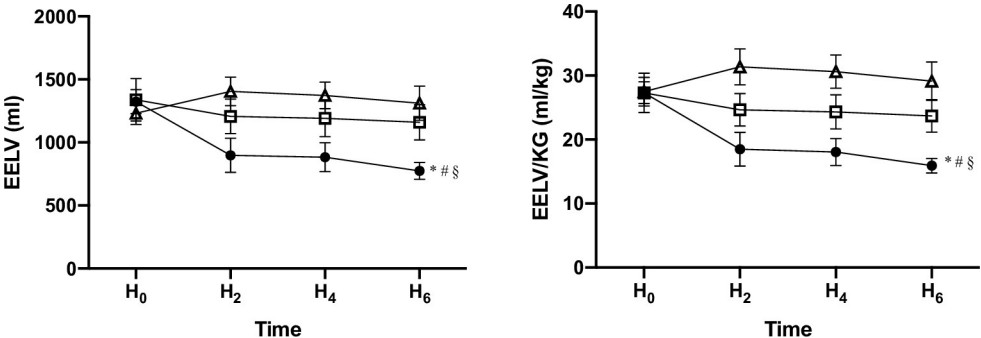

**Fig 2. Alterations of end-expiratory lung volume in absolute values and in relation to body weight in response to an intraabdominal pressure of 10 mmHg over 6 hours mechanical ventilation.** ● = group A with PEEP 5 cmH₂O, □ = group B with PEEP 10 cmH₂O ▲ = group C with PEEP 15 cmH₂O * = $p < 0.05$ HO vs. H6, # = $p < 0.05$ group A vs. group B, $ = $p < 0.05$ group A vs. group C. Mean values with SEM are illustrated.

$p < 0.05$) but not in group B (6 ± 1 vs. 7 ± 1 cmH₂O, $p > 0.05$). The $TTP_{EXP}$ stayed unaffected between H0 and H6 (Table 1) (S2 Fig).

Mean values with SEM are illustrated.

At H6 we found a lower EELV/kg in group A (16±1 ml/kg) compared to group B (24±3 ml/kg) and C (29±3 ml/kg) ($P < 0.05$) (Fig 2). Driving pressure decreased significantly with increasing PEEP (A: 17±1 cmH₂O, B: 15±1 cmH₂O, C: 12±1 cmH₂O, $p < 0.05$) (Fig 3).

Hemodynamic parameters and oxygenation are summarised in Table 1. We could not find a relevant alteration of cardiac index with higher PEEP nor was the P/F ratio significantly influenced.

### Lung injury score and wet-dry weight ratio

The global lung injury score was lowest in group B (0.17±0.02) compared to group A (0.30 ±0.04) and group C (0.32±0.02) ($p < 0.05$) (Fig 4). The wet-dry weight ratios were not different (S3 Fig).

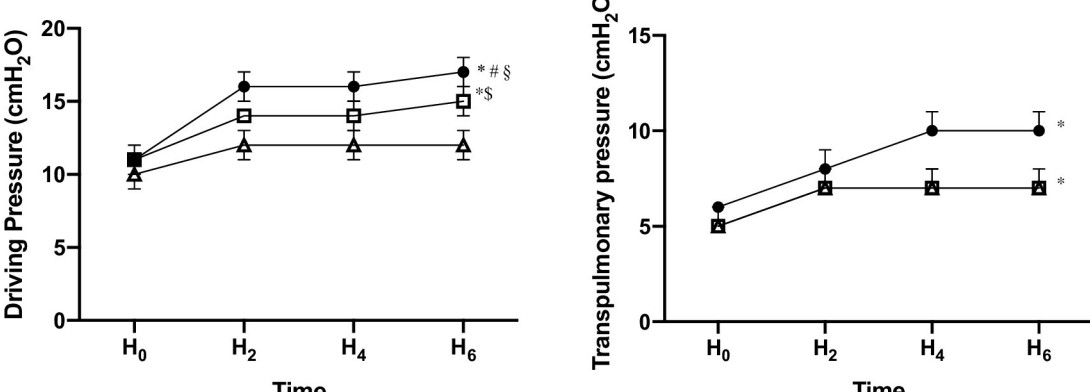

**Fig 3. Alterations of driving pressure and transpulmonary pressure in response to an intraabdominal pressure of 10 mmHg over 6 hours mechanical ventilation.** ● = group A with PEEP 5 cmH₂O, □ = group B with PEEP 10 cmH₂O ▲ = group C with PEEP 15 cmH₂O * = $p < 0.05$ HO vs. H6, # = $p < 0.05$ group A vs. group B, $ = $p < 0.05$ group A vs. group C, $ = $p < 0.05$ group B vs. group C. Mean values with SEM are illustrated.

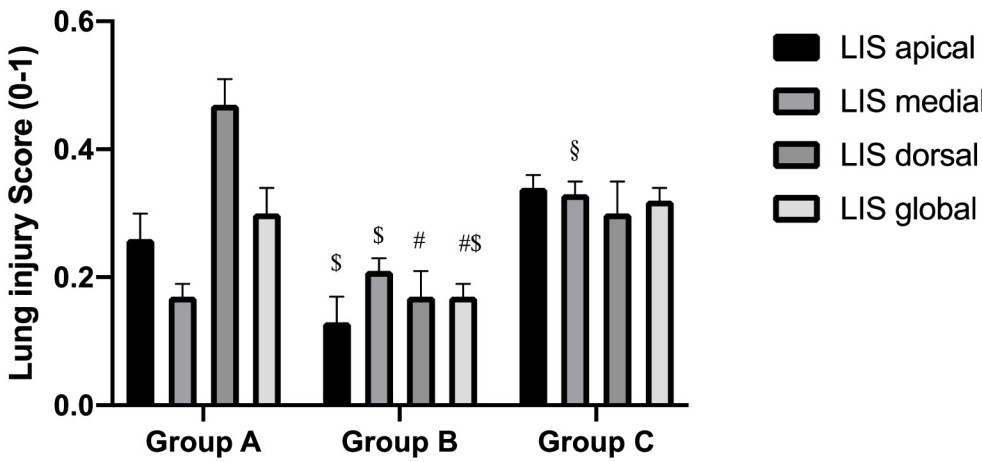

**Fig 4. Histologic assessment of lung injury.** Quantitative score for lung injury (from 0 = no damage to 1 = maximal alteration) calculated by the averaging score for five independent variables: neutrophils in the alveolar space, neutrophils in the interstitial space, hyaline membranes, proteinaceous debris filling the airspaces and alveolar septal thickening. Apical, medial, dorsal and global score (mean of scores for apical, medial and dorsal) are illustrated. # = p<0.05 group A vs. group B, § = p<0.05 group A vs. group C, $ = p<0.05 group B vs. group C. Mean values with SEM are illustrated.

## Discussion

### Main findings

The present animal study proved that even a moderate IAP of 10 mmHg caused changes in lung mechanics and a histological lung injury after 6 hours of ventilation. A PEEP of 10 cmH$_2$O overcame the intra-abdominal pressure induced alterations of EELV and revealed the lowest lung injury score.

### Alterations in lung mechanics

The main aim of our study was to describe the alterations in EELV, ΔP and ΔP$_L$ in a porcine model of moderately elevated IAP. These crucial parameters have an impact on the development or prevention of ventilator induced lung injury (VILI) [22, 23]. By implementing a water-filled balloon with an IAP of 10 mmHg for up to 6 hours, we detected a drop of EELV and EELV/kg bw with a PEEP of 5 cmH$_2$O over time. In the two groups with a PEEP of 10 and 15 cmH$_2$O, these reductions were not observed. P$_{Insp}$ did not rise proportionally to PEEP, which strongly suggests a regional recruitment due to an increasing PEEP. TPP$_{Insp}$ did not change significantly from H0 with a PEEP of 10 cmH$_2$O in contrast to the groups with PEEP 5 and 15 cmH$_2$O. The PEEP of 10 cmH$_2$O did not affect TPP$_{Exp}$ in significant manner. We interpret these results that the best compliance and lowest E$_L$ in this setting occurred with a PEEP of 10 cmH$_2$O. These results extend those obtained by Cortes-Puentes et al. [24], which were able to reveal an unresponsive TPP$_{Exp}$ with an increase of IAP. The authors argued that a rising IAP stiffens the chest wall, whereas aerated lung volume decreases, either due to derecruitment or a reduced stretch of air spaces that remain patent. Cortes-Puentes already showed that a PEEP of 10 cmH$_2$O is able to restore FRC, which appears to be reduced above an IAP of over 10 mmHg. Compared to our study design the researchers used intra-abdominal air insufflation and modified IAP levels between zero and 35 cmH$_2$O [24].

In our model with an IAP of 10 mmHg, E$_{RS}$ decreased with increasing PEEP, but E$_{CW}$ was the same in all groups. These findings are in line with studies in patients with ARDS. Krebs

et al. applied different PEEP levels (up to 20 cmH$_2$O) in 20 patients with ARDS; one half of the study population had IAH (with a mean IAP of 8 and 16 mmHg, respectively) [25]. PEEP was found to decrease E$_{RS}$ by decreasing E$_L$ without influencing E$_{CW}$ in both groups.

## PEEP in elevated IAP

In the context of elevated IAP, the management of PEEP is still a contentious issue [26]. The TPP$_{Exp}$ seems to be an important parameter. A negative TPP$_{Exp}$ should be avoided in order to prevent lung collapse and to overcome intratidal recruitment/derecruitment [27]. By inflating an intra-abdominal balloon, Regli et al. analyzed different PEEP levels (5, 8, 12 and 15 cmH$_2$O) unmatched to the level of IAP [28]. PEEP levels below the IAP were not able to prevent the decline of EELV. In a second study the same group now matched PEEP and IAP levels [29]. It must be noted that the authors found that EELV was preserved without any improvement in the P/F ratio. They argued that a reduction of cardiac output with higher PEEP levels is the main cause for the lack of improvement.

A human study in mechanically ventilated patients recently matched the PEEP to IAP. Only 10 of 18 patients tolerated this matched PEEP. In these 10 patients, the matched PEEP improved oxygenation but a PEEP matched 0.5 x IAP in cmH$_2$O did not [30]. In our study, TPP$_{Exp}$ was positive at all times and we were not able to find a relevant change in hemodynamic parameters. Neither could we find relevant alterations in P/F ratio between the groups or over time.

## Histological lung injury

Acute lung injury (ALI) in humans is characterized by disruption of the alveolar-capillary membrane barrier, proteinaceous alveolar exudate and pulmonary edema. We used the recommended scoring system for ALI in animal studies [21]. The lung injury score (LIS) was significantly lower in the group with a PEEP of 10 cmH$_2$O. PEEP of 10 cmH$_2$O was probably more lung protective during the ventilation than a lower or higher PEEP. As mentioned previously neither P/F ratio nor cardiac index were relevantly different between the groups.

Alteration in the wet-dry weight ratio is a typical feature of VILI, caused by high tidal volumes, endotoxin or bacterial induced ALI [31]. As expected, the groups did not differ regarding the wet-dry weight ratio. We therefore presume that the observed differences in LIS were not caused by lung edema.

## Experimental protocol

Some important aspects of our study differ from recent examinations and show the strength of the actual study.

1. We did not use an air inflated balloon to increase IAP. Instead we modified previously described methods to induce IAP [32, 33]. By installing water in a large 200-litre weather balloon, we aimed to simulate the leading cause of elevated IAP due to liquid ascites or oedematous tissue.

2. We only used a moderately elevated IAP of 10 mmHg and kept it for 6 hours. Since IAP is well known to be around 10 mmHg in critically ill patients [7, 10, 11] the investigation we present here therefore simulated a common scenario.

3. We tried to find the best PEEP in this setting. Therefore we matched PEEP and IAP levels based on the assumption that abdominal-thoracic transmission (ATT) is between 17 and 62% [3–6]. The ATT describes the percentage increase in thoracic pressures for each incremental increase of IAP.

## Limitations

This is an animal study. The results therefore cannot be transferred to human patients without any restrictions. We only used an IAP of 10mmHg. Hence, it is unclear to what extent the above-mentioned PEEP matching to IAP is useful and would be tolerated at higher IAP values. Clinical scenarios rarely end at 6 hours in human subjects and extending IAP and the study for a prolonged period may also reveal a difference in optimal PEEP recommendations.

## Conclusions

A moderately elevated IAP of 10 mmHg has already relevant effects on lung mechanics and on histological lung injury. Measuring bladder pressure should encourage the clinician to find the best PEEP in order to realize a safer ventilation strategy. In healthy porcine lungs with an IAP of 10 mmHg the application of a PEEP of 10 cmH$_2$O overcame the intra-abdominal pressure induced alterations of EELV. A PEEP of 10 cmH$_2$O revealed the lowest lung injury score.

## Supporting information

**S1 Checklist.**
(PDF)

**S1 Table. The lung injury scoring system.** Quantitative assessment for lung injury calculated by the averaging score for five independent variables and the resulting score (from 0 = no damage to 1 = maximal alteration).
(DOCX)

**S1 Fig. Alterations of static lung compliance to an intraabdominal pressure of 10 mmHg over 6 hours mechanical ventilation.** ● = group A with PEEP 5 cmH$_2$O, ⊞ = group B with PEEP 10 cmH$_2$O ▲ = group C with PEEP 15 cmH$_2$O * = p<0.05 HO vs. H6, # = p<0.05 group A vs. group B, $ = p<0.05 group A vs. group C, $ = p<0.05 group B vs. group C. Mean values with SEM are illustrated.
(DOCX)

**S2 Fig. Alterations of inspiratory transpulmonary pressure and expiratory transpulmonary pressure to an intraabdominal pressure of 10 mmHg over 6 hours mechanical ventilation.** ● = group A with PEEP 5 cmH$_2$O, ⊞ = group B with PEEP 10 cmH$_2$O ▲ = group C with PEEP 15 cmH$_2$O * = p<0.05 HO vs. H6. Mean values with SEM are illustrated.
(DOCX)

**S3 Fig. Wet-dry weight ratio of areas of the right lung.** Mean values with SEM are illustrated.
(DOCX)

**S1 Data.**
(XLSX)

## Acknowledgments

We thank Daniel Kazdal, Mark Kriegsmann and Katarina Kriegsmann, Institute of Pathology, Heidelberg University Hospital, Heidelberg, Germany for the contribution in dyeing and scanning of the histological lung slices and Christopher Buesch, Institute for medical biometry and informatic, Heidelberg University Hospital, Heidelberg, Germany, for statistical advices.

## Author Contributions

**Conceptualization:** Mascha O. Fiedler, Armin Kalenka.

**Data curation:** Mascha O. Fiedler, Alexander Harms, Thorsten Brenner, Armin Kalenka.

**Formal analysis:** Mascha O. Fiedler, B. Luise Deutsch, Dovile Diktanaite, Alexander Harms, Maik Brune, Florian Uhle, Armin Kalenka.

**Funding acquisition:** Armin Kalenka.

**Investigation:** Mascha O. Fiedler, B. Luise Deutsch, Emilis Simeliunas, Maik Brune, Armin Kalenka.

**Methodology:** B. Luise Deutsch, Alexander Harms, Maik Brune, Armin Kalenka.

**Project administration:** Mascha O. Fiedler, Armin Kalenka.

**Resources:** Markus Weigand, Thorsten Brenner, Armin Kalenka.

**Software:** B. Luise Deutsch, Armin Kalenka.

**Supervision:** Armin Kalenka.

**Validation:** Maik Brune, Armin Kalenka.

**Visualization:** Alexander Harms, Armin Kalenka.

**Writing – original draft:** Mascha O. Fiedler, Armin Kalenka.

**Writing – review & editing:** Mascha O. Fiedler, B. Luise Deutsch, Emilis Simeliunas, Dovile Diktanaite, Alexander Harms, Maik Brune, Florian Uhle, Markus Weigand, Thorsten Brenner, Armin Kalenka.

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
