## [Decision Letter · Decision Letter 0]

3 Jan 2020

PONE-D-19-29054

Effect of moderate elevated intra-abdominal pressure on lung mechanics and histological lung injury at different positive end-expiratory pressures

PLOS ONE

Dear Dr. Kalenka,

Thank you for submitting your manuscript to PLOS ONE. After careful consideration, we feel that it has merit but does not fully meet PLOS ONE’s publication criteria as it currently stands. Therefore, we invite you to submit a revised version of the manuscript that addresses the points raised during the review process.

The authors are to be commended for the design and reporting of a study relevant to clinical practice in the critically ill patient population.  Although the manuscript is generally well written and well presented, there are several errors of English language usage and a few typographical errors that should be corrected.  These errors were identified during a careful line-by-line review of the manuscript.  The assistance of a copy editor with expertise in English publication may prove useful. 

More significantly, there are several errors in the references.  First, and most importantly, the references are to be cited in order of mention in the manuscript.  For example, the first references mentioned in the manuscript are Reference 5 and Reference 6.  Therefore, Reference 5 should actually be Reference 1.  Accordingly, all of the references will have to be re-ordered and re-numbered.  Additionally, several errors in the formatting of the references must be corrected.  Some of the titles of the references are capitalized when they should be listed in lower case; some of the terminal page numbers are missing; and there are discrepancies of years of publication in a few of the references (two different years of publication listed within the same reference).  The authors are directed to the PLOS ONE submission guidelines with respect to formatting of references:  “Because all references will be linked electronically as much as possible to the papers they cite, proper formatting of the references is crucial.”

A revision that focuses on attention to these details and correction of the errors is required for the manuscript to be considered for publication in PLOS ONE.   

We would appreciate receiving your revised manuscript within 45 days of the date of this letter. To enhance the reproducibility of your results, we recommend that if applicable you deposit your laboratory protocols in protocols.io, where a protocol can be assigned its own identifier (DOI) such that it can be cited independently in the future. For instructions see: http://journals.plos.org/plosone/s/submission-guidelines#loc-laboratory-protocols

We look forward to receiving your revised manuscript.

Kind regards,

Linda L. Maerz, MD

Academic Editor

PLOS ONE

Journal Requirements:

2. Please include further information regarding your in vivo study, per our guidelines (http://journals.plos.org/plosone/s/submission-guidelines#loc-animal-research).

Specifically, please provide details regarding:

- The number and type of animals used

- The source of the animals

- Animal health monitoring, including frequency and criteria and any efforts made to reduce suffering and distress, such as administering analgesics

- whether humane endpoints were in place during the study and how they were applied

- the method of anesthesia and euthanasia used

-  any mortality that occurred outside of planned euthanasia or humane endpoints

In addition please complete and submit a copy of the ARRIVE Guidelines checklist, a document that aims to improve experimental reporting and reproducibility of animal studies for purposes of post-publication data analysis and reproducibility: https://www.nc3rs.org.uk/arrive-guidelines. Please include your completed checklist as a Supporting Information file. Note that if your paper is accepted for publication, this checklist will be published as part of your article.

We thank you for your attention to these requests.

4. Please include your tables as part of your main manuscript and remove the individual files.

Please note that supplementary tables should be uploaded as separate "supporting information" files.

5. Please upload a copy of Figure 5, to which you refer in your text on page 11. If the figure is no longer to be included as part of the submission please remove all reference to it within the text.

Reviewers' comments:

Reviewer's Responses to Questions

**Comments to the Author**

1. Is the manuscript technically sound, and do the data support the conclusions?

Reviewer #1: Yes

Reviewer #2: Yes

2. Has the statistical analysis been performed appropriately and rigorously? 

Reviewer #1: Yes

Reviewer #2: Yes

3. Have the authors made all data underlying the findings in their manuscript fully available?

Reviewer #1: Yes

Reviewer #2: Yes

4. Is the manuscript presented in an intelligible fashion and written in standard English?

Reviewer #1: Yes

Reviewer #2: Yes

5. Review Comments to the Author

Reviewer #1: Important question to answer and appropriate porcine model. Technically sound with reasonable statistical analysis and available data. It seems as though selection of an intra-abdominal study pressure of 10 mm Hg was thoughtful. Could consider follow up study at pressures of 15 mm Hg and perhaps 20 mm Hg to evaluate optimal PEEP in those instances. Clinical scenarios rarely end at 6 hrs in human subjects and extending IAP and the study for a prolonged period may also reveal a difference in optimal PEEP recommendations for short term vs. long term intubation.

Reviewer #2: This manuscript is a well-written summary and discussion of a well-designed study. It is a small study and an animal study, and so further study will be needed prior to any significant change in practice. However, the basics of the lung mechanics are relevant and have not been well-established in patients with mildly elevated intra-abdominal pressure. Importantly, this is not meant to comment on patients with abdominal compartment syndrome or larger increases in IAP.

I have no specific concerns about the ethics or publication of this study.

I recommend this study for publication.

6. PLOS authors have the option to publish the peer review history of their article (what does this mean?). If published, this will include your full peer review and any attached files.

Reviewer #1: No

Reviewer #2: No

---

## [Author Response · Author response to Decision Letter 0]

4 Jan 2020

The respond to reviewers are mentioned in the file "respond to reviewers"

---

## [Editor Report · Decision Letter 1]

22 Jan 2020

PONE-D-19-29054R1

Effect of moderate elevated intra-abdominal pressure on lung mechanics and histological lung injury at different positive end-expiratory pressures

PLOS ONE

Dear Dr. Kalenka,

Thank you for submitting your manuscript to PLOS ONE. After careful consideration, we feel that it has merit but does not fully meet PLOS ONE’s publication criteria as it currently stands. Therefore, we invite you to submit a revised version of the manuscript that addresses the points raised during the review process.

The authors are to be commended for beginning to address some of the technical concerns raised regarding their manuscript.  However, some of the English usage and typographical errors persist, and some new errors are identified in the revised manuscript.  For example:

In the templated abstract (at the very beginning of the submission) formatting irregularities that were not present previously occur with some of the abbreviations.Also in the templated abstract, the words “Methods,” “Results,” and “Conclusions” should appear at the beginning of the paragraphs to which they refer.Page 5, line 12:Is “0,3 mg/kg” correct, or should it be “0.3” or “0-3” or something else?Page 5, line 14:Is “3,6 mg/kg” correct, or should it be something else (similar to the prior item)?Page 6, line 1:“hemodynamic” should be “hemodynamically”Page 7, line 24: “weighted” should be “weighed”Page 8, lines 4-7:The two sentences beginning with “In case of significant results . . . “ are awkwardly worded and difficult to understand.Page 9, line 19: “to” should be “from”Page 10, line 2:“ration” should be “ratio”Page 11, lines 21-24:The two sentences beginning with “The authors argued that. . . “ should be one sentence with a comma after “decreases” in line 23, or should be reworded altogether.Page 13, line 3:“exsudate” should be “exudate”Page 13, line 4:Use “ALI” instead of “acute lung injury” since the acronym has already been introduced.Page 13, line 9:“Alterations” should be “Alteration”Page 21, line 9:“hold” should be “held”Page 21, line 12:“bodyweight” should be “body weight”Figure 1:“FiO2 0,4” should be “FiO2 0.4” if the 0.4 parameter is correct.However, this is contradicted in the manuscript:Page 5, line 20 states “(FiO2) of 0.3”Figure 1 legend, last line:“hold” should be “held”

Additionally, new errors are noted in the citing and reporting of the references.  Specifically:

References 25 and 26 are cited on page 11, line 11.However, there is no antecedent citing of references 22, 23, or 24.Reference 24 is cited out of order (after the aforementioned citing of References 25 and 26).Reference 22 and Reference 23 are not cited at all.The last reference cited in the manuscript is Reference 30.However, 33 references are listed at the end of the manuscript.The authors did not provide tracked changes for their revisions to the list of references at the end of the manuscript.Reference 5:The acronym “acs” should be “ACS”Reference 21:“american thoracic society” should be “American Thoracic Society”Reference 27:The title of the manuscript should be lower case.

The above notations are examples of errors.  Again, the assistance of a copy editor with expertise in English publication may prove useful.  An additional revision that focuses attention on correction of these and any additional errors is required for the manuscript to be considered for publication in PLOS ONE.  Of note, the errors in reference order and citation are particularly important.

We would appreciate receiving your revised manuscript by 45 days from the date of this letter. To enhance the reproducibility of your results, we recommend that if applicable you deposit your laboratory protocols in protocols.io, where a protocol can be assigned its own identifier (DOI) such that it can be cited independently in the future. For instructions see: http://journals.plos.org/plosone/s/submission-guidelines#loc-laboratory-protocols

We look forward to receiving your revised manuscript.

Kind regards,

Linda L. Maerz, MD

Academic Editor

PLOS ONE

---

## [Author Response · Author response to Decision Letter 1]

29 Jan 2020

There were no specific points by the reviewers for our revised manuscript.

---

## [Editor Report · Decision Letter 2]

18 Feb 2020

PONE-D-19-29054R2

Effect of moderate elevated intra-abdominal pressure on lung mechanics and histological lung injury at different positive end-expiratory pressures

Dear Dr. Kalenka, 

Thank you for submitting your revised manuscript to PLOS ONE. After careful consideration, we feel that it has merit but does not fully meet PLOS ONE’s publication criteria as it currently stands. Therefore, we invite you to again submit a revised version of the manuscript that addresses the points raised during the review process.

The authors are to be commended for addressing the majority of the requested revisions.  In particular, the re-numbering of the references is appreciated.  However, a few items remain to be addressed:

In the templated abstract (at the very beginning of the submission) formatting irregularities that were not present in the original submission occur with some of the abbreviations.  Specifically, this pertains to the abbreviations H2O, Cstat, and ΔPL.Figure 1:  "FiO2 0,4" should be "FiO2 0.4"Figure 1 legend, last line: “hold” should be “held”Reference 27:The title of the manuscript should be lower case.

The authors’ attention to these remaining details will be appreciated.

We would appreciate receiving your revised manuscript within 45 days of the date of this letter. To enhance the reproducibility of your results, we recommend that if applicable you deposit your laboratory protocols in protocols.io, where a protocol can be assigned its own identifier (DOI) such that it can be cited independently in the future. For instructions see: http://journals.plos.org/plosone/s/submission-guidelines#loc-laboratory-protocols

We look forward to receiving your revised manuscript.

Kind regards,

Linda L. Maerz, MD

Academic Editor

PLOS ONE

---

## [Author Response · Author response to Decision Letter 2]

23 Feb 2020

In our third revision is no need to response to the reviewers.

---

## [Editor Report · Decision Letter 3]

10 Mar 2020

Effect of moderate elevated intra-abdominal pressure on lung mechanics and histological lung injury at different positive end-expiratory pressures

PONE-D-19-29054R3

Dear Dr. Kalenka,

We are pleased to inform you that your manuscript has been judged scientifically suitable for publication and will be formally accepted for publication once it complies with all outstanding technical requirements.

With kind regards,

Linda L. Maerz, MD

Academic Editor

PLOS ONE

---

## [Editor Report · Acceptance letter]

30 Mar 2020

PONE-D-19-29054R3 

Effect of moderate elevated intra-abdominal pressure on lung mechanics and histological lung injury at different positive end-expiratory pressures 

Dear Dr. Kalenka:

I am pleased to inform you that your manuscript has been deemed suitable for publication in PLOS ONE. Congratulations! Your manuscript is now with our production department. 

With kind regards,

on behalf of

Dr. Linda L. Maerz 

Academic Editor

PLOS ONE